# Natural Products Targeting BCR-ABL: A Plant-Based Approach to Chronic Myeloid Leukemia Treatment

**DOI:** 10.3390/molecules30214160

**Published:** 2025-10-22

**Authors:** Louisa Pechlivani, Alexandros Giannakis, Chrissa Sioka, Georgios A. Alexiou, Athanassios P. Kyritsis

**Affiliations:** 1Neurosurgical Institute, University of Ioannina, 45500 Ioannina, Greece; louisapechlivani@gmail.com (L.P.); csioka@yahoo.com (C.S.); galexiou@uoi.gr (G.A.A.); 2Department of Neurology, Faculty of Medicine, School of Health Sciences, University of Ioannina, 45500 Ioannina, Greece; papadates@gmail.com; 3Nuclear Medicine, Faculty of Medicine, School of Health Sciences, University of Ioannina, 45500 Ioannina, Greece; 4Department of Neurosurgery, University of Ioannina, 45500 Ioannina, Greece

**Keywords:** BCR-ABL inhibition, chronic myeloid leukemia, natural products, tyrosine kinase inhibitor resistance, plant-derived compounds

## Abstract

The BCR-ABL fusion oncoprotein, a constitutively active tyrosine kinase, plays a central role in the pathogenesis of chronic myeloid leukemia (CML). While tyrosine kinase inhibitors (TKIs) have transformed CML treatment, issues such as drug resistance, particularly involving mutations like T315I, and adverse effects underscore the need for alternative or complementary therapeutic strategies. Natural products derived from plants have long served as a reservoir for anticancer agents, offering structural diversity and multi-targeted bioactivity. Notably, many plant-based compounds exhibit anticancer effects with comparatively lower toxicity and fewer side effects than synthetic TKIs, making them attractive candidates for safer long-term use. This review explores the recent advances in plant-based natural compounds that directly or indirectly inhibit BCR-ABL kinase activity and its downstream signaling pathways. Key compounds are discussed with respect to their mechanisms of action, structure–activity relationships, and potential to overcome TKI resistance. Several of these compounds directly target BCR-ABL or promote its degradation, while others inhibit downstream effectors such as STAT5 and PI3K/Akt, leading to apoptosis and growth inhibition of leukemic cells. The synergistic potential of these natural products with existing TKIs and their promise to target drug-resistant CML cells further highlight their translational value. By integrating insights from molecular pharmacology, medicinal chemistry, and leukemia biology, this review supports the continued investigation of plant-derived agents as novel or adjunctive therapies against BCR-ABL-driven leukemias.

## 1. Introduction

Chronic myeloid leukemia (CML) is a myeloproliferative neoplasm arising from hematopoietic stem cells and characterized by excessive proliferation of myeloid lineage cells in bone marrow and peripheral blood. It accounts for approximately 15–20% of adult leukemias and has an annual incidence of ~0.8–1.5 per 100,000 globally, with a slight incidence preference in men. Clinically, most patients present in a chronic phase, which may progress, if untreated, to an accelerated phase and eventually a blast crisis [1,2]. CML is marked by the presence of the Philadelphia (Ph) chromosome, a product of reciprocal translocation t (9;22). This cytogenetic aberration fuses the 5′ segment of the BCR gene on chromosome 22 with the 3′ segment of the ABL gene on chromosome 9, resulting in the formation of the BCR-ABL fusion gene. The translated BCR-ABL oncoprotein exhibits constitutive tyrosine kinase activity, which activates multiple signaling cascades involved in cell proliferation, survival, and genomic instability, driving the leukemogenic process in CML (Figure 1) [3,4].

Crystallographic studies of the BCR-ABL oncoprotein have revealed a conserved catalytic core composed of two structural domains, an N-terminal lobe (N-lobe), primarily consisting of β-sheets, and a C-terminal lobe (C-lobe), predominantly formed by α-helices. The ATP-binding cleft is located at the interface of these lobes, serving as the functional site for kinase activity [3]. There are four major BCR-ABL isoforms, P185, P190, P210, and P230, which arise from alternative breakpoints within the BCR gene. These isoforms are associated with different leukemic phenotypes. P185 and P190 are commonly found in acute lymphoblastic leukemia (ALL), P210 appears in most CML cases, and P230 is associated with chronic myelomonocytic leukemia (CMML) [5]. Functionally, BCR-ABL acts as an active tyrosine kinase, utilizing ATP to phosphorylate downstream substrates. This activity activates multiple signaling pathways, such as RAS/MAPK, PI3K/Akt, and JAK/STAT, leading to uncontrolled cell proliferation, resistance to apoptosis, and altered cell adhesion (Figure 2). These insights into the molecular mechanisms of BCR-ABL-driven oncogenesis catalyzed the development of tyrosine kinase inhibitors (TKIs) as targeted therapies [6].

TKIs exert their therapeutic effect by competitively binding to the ATP-binding site of BCR-ABL, thereby blocking its kinase activity. Imatinib, the first clinically approved TKI, revolutionized the treatment landscape of CML. However, the emergence of resistance mechanisms, such as kinase domain mutations, like the T315I, BCR-ABL overexpression, and activation of bypass pathways, necessitated the development of next-generation inhibitors. Structural comparison between the wild-type and T315I-mutant forms of BCR-ABL reveals key alterations in the ATP-binding site. The threonine-to-isoleucine substitution at position 315 disrupts hydrogen bonding with many TKIs and introduces steric hindrance, explaining resistance to earlier inhibitors (Figure 3) [7,8].

To address these challenges, second-generation TKIs (nilotinib, dasatinib, bosutinib), third-generation TKIs (ponatinib, olverembatinib, vodobatinib), and, more recently, fourth-generation TKIs (PF-114) and inhibitors that specifically target the ABL Myristoyl Pocket (STAMP) (asciminib) have been introduced (Figure 4). These agents demonstrate enhanced potency, broader mutation coverage, and improved pharmacologic profiles, though long-term use remains limited by toxicity and cost [7,8].

Each generation of TKIs brings its own set of side effects, which can impact patient quality of life and even lead to treatment changes. Common adverse effects include fatigue, gastrointestinal symptoms, and skin reactions, but more serious complications such as cardiovascular events, liver toxicity, and metabolic disturbances have also been reported. Some newer TKIs, though more effective against resistant mutations, may carry increased risk for arterial occlusive events or pancreatitis. As treatment often continues for years, managing these safety concerns has become a crucial part of long-term care in CML [9].

The objective of this review is to provide a comprehensive overview of plant-derived natural products targeting BCR-ABL in CML. We discuss their molecular mechanisms, potential to circumvent resistance, and promise as safer or adjunct therapies in CML.

## 2. Tumor Resistance to Therapy

Chemotherapy remains a primary treatment for cancer, although the molecular mechanisms responsible for drug sensitivity or resistance in different tumor types are still not fully understood. Selecting the optimal chemotherapy regimen for a given cancer type is quite challenging and can lead to very different outcomes for each patient. Most chemotherapeutic agents cause DNA damage and activate complex signaling networks, leading to cell cycle arrest or apoptosis. During oncogenesis and tumor progression, almost all cancer cells disrupt the function of the DNA damage response (DDR) pathway. DDR disruptions result in genomic instability, which could contribute to drug resistance. Therefore, chemotherapy effectiveness may depend on differences in DDR function between normal and cancer cells [10,11].

Cancer cells can evade chemotherapy through various mechanisms involving altered regulation of the cell cycle, apoptosis, and cell adhesion. Resistance to TKIs, whether primary or acquired, is one of the major challenges in cancer treatment. The main resistance mechanisms are classified into two categories, BCR-ABL-dependent and BCR-ABL-independent (Figure 5). BCR-ABL-dependent resistance is mainly influenced by the site of mutations in BCR-ABL and its uncontrolled gene expression. One of the most common resistance mechanisms is a mutation in the ABL kinase domain, which directly or indirectly prevents TKI binding, either by altering BCR-ABL conformation, reducing its affinity for the inhibitor, or by obstructing the binding site. Point mutations in the ABL kinase domain are observed in over 50% of CML patients who develop resistance, especially in those with acquired resistance rather than primary. Such mutations are associated with a higher risk of developing additional genetic abnormalities, leading to a worse prognosis and faster disease progression [12,13,14].

For example, patients resistant to imatinib were more likely to relapse due to the emergence of new mutations, compared to those without such mutations. The likelihood of mutations increases with disease progression, and overexpression of the BCR-ABL kinase also plays a significant role by promoting the self-renewal of leukemic stem cells. To date, over 100-point mutations have been confirmed, with the most common and deadly being T315I, where threonine is replaced by isoleucine at position 315 of BCR-ABL. This mutation renders nilotinib ineffective. Thus, BCR-ABL kinase is a highly significant therapeutic target in cancer [15,16,17].

BCR-ABL-independent resistance mechanisms are mainly influenced by increased drug efflux, reduced drug uptake, and selective activation of tumor-promoting pathways. Drug effectiveness depends on bioavailability and the ability to reach the pharmacological target. Transporter proteins significantly impact the intracellular concentration of drugs. Resistance can arise from reduced influx or, conversely, excessive efflux of TKIs. Low intracellular availability of a TKI leads to weaker apoptotic activity, favoring the development of mutations that drive drug resistance. A frequent mechanism of TKI resistance involves activation of alternative signaling pathways, such as PI3K/Akt, JAK/STAT, RAS/MAPK, and SRC, which compensate for BCR-ABL inhibition. As a result, cancer cells continue to proliferate despite BCR-ABL suppression and clinically, this means that some CML patients may be resistant to all available TKIs [18,19].

TKIs are metabolized in the liver, often causing hepatotoxicity, and their bioavailability can be influenced by the metabolism of other drugs commonly administered to these patients (Table 1). All co-administered medications should be evaluated for potential interactions, especially in patients with multiple comorbidities or poor therapeutic response [20].

These limitations have prompted exploration of alternative or adjunctive therapeutic strategies. Natural products from plants offer a rich source of chemical diversity and multifunctional bioactivity (Figure 6). Many exhibit lower systemic toxicity and fewer off-target effects compared to conventional TKIs, making them attractive candidates for chronic or combinatorial therapy. Some plant-derived molecules demonstrate the ability to directly inhibit BCR-ABL, induce its degradation, or suppress oncogenic signaling, often with synergistic potential when combined with TKIs. Accordingly, these natural compounds can act either as direct BCR-ABL inhibitors, which block the kinase activity of the oncoprotein, or as indirect modulators that promote its degradation or suppress downstream signaling, highlighting their potential to complement conventional TKI therapy.

## 3. Direct BCR-ABL Kinase Inhibitors

### 3.1. Emodin

Emodin (1,3,8-trihydroxy-6-methylanthraquinone) is a natural compound found in plants like rhubarb (*Rheum palmatum* L.), *Polygonum cuspidatum* and *Polygonum multiflorum* [29]. It has shown promising effects against CML, especially by targeting the BCR-ABL oncoprotein and several studies have explored how emodin works in CML cells, including those that are resistant to commonly used TKIs like imatinib. Emodin helps reverse multidrug resistance by competitively inhibiting P-glycoprotein and lowering its expression, thereby increasing intracellular drug accumulation and potentially restoring the efficacy of agents like imatinib [30]. It also reduces BCR-ABL expression in K562 cells (a CML model cell line) and modulates the PTEN/PI3K/Akt signaling pathway, promoting apoptosis and impairing cell survival [31]. Novel derivatives of emodin have demonstrated inhibitory effects on both wild-type and T315I mutant BCR-ABL cells, significantly downregulating BCR-ABL and its downstream pathways, which supports their potential in overcoming drug resistance [32,33].

Emodin in co-administration with imatinib has been shown to improve therapeutic response in CML cells by targeting multiple resistance mechanisms. It enhances sensitivity to imatinib through downregulation of BCR-ABL and STAT5, while also exerting an allosteric inhibitory effect on BCR-ABL kinase activity [34]. Furthermore, when combined with AZT (3′-azido-3′-deoxythymidine), an inhibitor used as treatment for HIV and malignancies like leukemia, has been shown to enhance the inhibition of proliferation and promote apoptosis in K562 leukemia cells. This synergistic effect is associated with the regulation of the EGR1 (early growth response protein-1) and Wnt/β-catenin signaling pathways, which are implicated in leukemic cell survival and drug resistance [35].

Emodin suffers from poor oral bioavailability, estimated at around 3% in animal models. This low absorption is primarily due to extensive glucuronidation and rapid first-pass metabolism in the liver and intestines, with much of the unchanged drug and its metabolites eliminated via feces and urine [36]. On the safety front, emodin is generally well tolerated at moderate doses in preclinical models. However, long-term exposure or high dosages have been linked to hepatotoxicity, renal injury, and reproductive toxicity in animal studies [37,38]. These findings underscore the need for careful dose optimization and highlight the potential benefits of enhanced drug delivery approaches to minimize risks while improving therapeutic efficacy.

### 3.2. Oridonin

Oridonin, a natural diterpenoid compound from *Rabdosia rubescens*, has demonstrated notable inhibitory effects on BCR-ABL by promoting its degradation and thereby diminishing its oncogenic activity. This action interferes with leukemia cell growth and positions oridonin as a promising candidate for improving therapeutic strategies against BCR-ABL-positive malignancies. Oridonin’s mechanism involves reducing BCR-ABL protein levels through a chaperone-mediated proteasomal degradation pathway. In this process, molecular chaperones guide the recognition of BCR-ABL and facilitate its breakdown by the proteasome. By destabilizing the BCR-ABL protein rather than merely inhibiting its kinase activity, oridonin effectively disrupts downstream signaling essential for leukemic cell survival [39,40]. Additionally, oridonin has been found to inhibit the growth of both imatinib-sensitive and imatinib-resistant K562 cells by reducing BCR-ABL expression and inducing apoptosis. This effect is associated with interference in BCR-ABL-related signaling pathways, leading to the suppression of leukemic cell proliferation [41]. Specifically, oridonin inhibits BCR-ABL activity in leukemia cells by binding to the cysteine-153 residue within the HSF1 domain, leading to increased expression of HSP70 and ubiquitin proteins that promote BCR-ABL degradation. In imatinib-resistant K562 cells, oridonin suppresses proliferation by downregulating Bcl-2 and phosphorylated Lyn, a kinase linked to resistance through the mTOR pathway. These findings suggest that oridonin may overcome resistance to BCR-ABL-targeted therapies and potentially act in a broader, non-specific manner against TKI resistance [42]. Interestingly, oridonin has been shown to work synergistically with imatinib to inhibit the growth of Philadelphia chromosome-positive acute lymphoblastic leukemia cells. This combination effectively suppresses the activation of the LYN/mTOR signaling pathway, which is involved in cell survival and proliferation. By targeting this pathway, oridonin helps enhance the anti-leukemic effects of imatinib, acting as a potential supportive treatment in BCR-ABL-positive leukemia [43].

Jiyuan Oridonin A, a derivative of oridonin, has been shown to promote differentiation in imatinib-resistant CML cells carrying the BCR-ABL T315I mutation. This compound reduces BCR-ABL protein levels and disrupts its downstream signaling, leading to decreased proliferation and increased maturation of leukemic cells, thus, indicating its potential as a therapeutic approach for resistant CML cases [44].

While oridonin is a compound with promising anticancer properties, its oral bioavailability is quite low, typically only around 4–5% in animal studies, due to poor water solubility and significant first-pass metabolism. While generally well-tolerated, some preclinical studies have shown that high doses may lead to toxicity, such as appetite loss and liver or kidney damage in rodents [45,46]. These safety concerns highlight the importance of dosage optimization and lead to ongoing efforts in the development of derivatives and sophisticated delivery systems, including PEGylated analogs, cyclodextrin complexes, and nanocarriers, to improve solubility, enhance biological activity, and reduce toxicity [47,48].

### 3.3. Berberine

Berberine, a natural isoquinoline alkaloid derived from *Berberis vulgaris*, has demonstrated the ability to degrade both wild-type and T315I mutant forms of BCR-ABL through an autophagy-dependent mechanism [49]. It has been shown to inhibit the growth of K562 leukemia cells by inducing both differentiation and apoptosis and its action is associated with the downregulation of BCR-ABL expression [50]. This effect is mediated by the recruitment of the E3 ubiquitin ligase LRSAM1, which facilitates ubiquitination and lysosomal degradation of BCR-ABL, thereby reducing its protein stability rather than directly inhibiting kinase activity. By lowering BCR-ABL levels, berberine disrupts oncogenic signaling pathways essential for leukemic cell survival, leading to suppressed proliferation, induction of differentiation, and increased apoptosis. These effects suggest that berberine may serve as a potential therapeutic agent in targeting BCR-ABL-positive leukemias. Importantly, this mechanism remains active in imatinib-resistant cells, suggesting its potential to overcome TKI resistance [51,52]. Berberine in co-administration with imatinib has shown a synergistic anti-leukemic effect in CML cells. This combination promotes autophagic degradation of BCR-ABL, including the T315I mutant, leading to enhanced apoptosis and inhibition of proliferation in both imatinib-sensitive and -resistant CML models. Additionally, berberine alleviates imatinib-induced cardiotoxicity by inhibiting Nrf2-dependent ferroptosis, suggesting that it not only potentiates TKI efficacy but also may reduce associated side effects [51,53].

Nevertheless, Berberine’s clinical use faces some important challenges. One of the biggest issues is poor oral bioavailability. When taken by mouth, only a small fraction of berberine reaches the bloodstream because it is poorly absorbed, rapidly metabolized, and actively pumped out of intestinal cells. To address this limitation, new formulations of berberine are being developed, and some have already shown improved absorption in both laboratory models and early human studies. From a safety perspective, berberine is generally well tolerated, with the most common side effects being mild gastrointestinal discomfort such as stomach upset or diarrhea [54,55]. However, because it interacts with multiple cellular pathways, there is a risk of drug interactions and off-target effects [56]. This is especially important in CML, where patients are often on long-term TKI therapy, and potential interactions could affect treatment outcomes. Overall, while berberine shows promise, more research is needed to determine whether its pharmacological benefits can be translated into safe and effective use alongside TKIs in clinical practice.

### 3.4. Chlorogenic Acid

Chlorogenic acid, a naturally occurring polyphenol found in coffee, fruits, and vegetables, has been studied for its ability to target oncogenic signaling in CML. One of its key effects is the inhibition of BCR-ABL tyrosine kinase activity. Early studies showed that chlorogenic acid suppresses BCR-ABL phosphorylation and blocks its downstream signaling, ultimately impairing the survival of leukemic cells. This inhibition is accompanied by the activation of p38 MAPK, which plays a central role in initiating apoptosis in BCR-ABL-positive cells. Importantly, chlorogenic acid not only interferes with BCR-ABL activity but also selectively induces apoptosis in leukemic cells without significantly affecting BCR-ABL-negative cells, suggesting a degree of therapeutic specificity [57]. Further investigations revealed that reactive oxygen species (ROS) are critical mediators of chlorogenic acid’s pro-apoptotic effect. Chlorogenic acid treatment increases intracellular ROS levels, which contributes to mitochondrial dysfunction, loss of membrane potential, and activation of the caspase cascade. Blocking ROS generation has been shown to attenuate chlorogenic acid-induced apoptosis, highlighting the importance of oxidative stress in its mechanism of action. These dual actions, direct inhibition of BCR-ABL kinase and ROS-dependent induction of apoptosis, make chlorogenic acid a promising candidate for targeting both kinase-driven proliferation and survival pathways in CML cells [58].

However, despite these encouraging findings, research on chlorogenic acid in leukemia remains largely preclinical. Additional studies are needed to clarify its pharmacokinetics, safety, and potential synergy with existing TKIs before translation into clinical applications. Chlorogenic acid demonstrates low oral bioavailability, largely because it is rapidly absorbed and metabolized, with much of the compound transformed by the gut microbiota and eliminated before reaching systemic circulation [59,60]. In clinical studies, only about 8% of the ingested dose or its metabolites is recovered in urine over a 24 h period, underscoring its limited systemic exposure [61]. Regarding safety, chlorogenic acid generally shows a favorable tolerability profile in humans, with no significant adverse effects reported at standard dietary levels. However, some preclinical evidence in animal models suggests that high doses may impart mild liver biochemical changes, highlighting the need for caution and further safety evaluations [62,63].

### 3.5. β-Phenylethyl Isothiocyanate

β-Phenylethyl isothiocyanate (PEITC), a naturally occurring compound found in cruciferous vegetables, has demonstrated significant inhibitory effects on BCR-ABL and its downstream signaling in CML. Studies have shown that PEITC can effectively target both imatinib-sensitive and imatinib-resistant CML cells, including those harboring the T315I mutation, which is notably resistant to conventional TKIs [64]. One of the primary mechanisms involves the induction of oxidative stress through the generation of ROS, leading to mitochondrial damage, activation of caspases, and apoptosis in leukemic cells [65]. By altering the redox balance, PEITC triggers cell death pathways that bypass the reliance on BCR-ABL kinase inhibition, offering a potential strategy for overcoming drug resistance. In addition to its redox activity, PEITC has been reported to interfere with protein homeostasis by inhibiting deubiquitinating enzymes, resulting in the accumulation of ubiquitinated proteins and disruption of cellular survival mechanisms [66]. Importantly, PEITC in co-administration with imatinib has demonstrated a synergistic anti-leukemic effect by inhibiting the crosstalk between BCR-ABL and protein kinase C (PKC) signaling, pathways that cooperate to maintain leukemic cell growth and survival. By disrupting this interaction, PEITC enhances the sensitivity of CML cells to imatinib, suggesting that it may be useful in combination therapy [67]. Collectively, these findings highlight that PEITC acts through multiple mechanisms, including ROS generation, proteostasis disruption, and inhibition of critical signaling crosstalk, to suppress CML cell growth. Further preclinical and translational studies are warranted to explore its potential as either a standalone or adjunct treatment strategy in targeting BCR-ABL-driven leukemia.

Interestingly, PEITC shows impressive oral bioavailability, with animal studies indicating absorption rates exceeding 90–100%, likely due to its stability and low clearance in circulation [68]. In human studies, dietary doses of PEITC up to approximately 40 mg per day are considered safe, though higher intakes, between 120 and 160 mg/day, have been linked to mild toxicity and potential interactions with other drugs [69]. Clinical data in other settings suggest that PEITC-fortified nutritional supplements can be well tolerated over multi-week periods, with only mild, transient gastrointestinal side effects (nausea, dry mouth) observed, and no serious adverse events reported [70]. Overall, PEITC appears bioavailable and tolerable at moderate doses, but its broad biological activity and possible drug interactions merit caution and further study in future clinical contexts, especially when considering long-term use or combination with other medications.

### 3.6. Gallic Acid

Gallic acid, a naturally occurring polyphenolic compound, has attracted considerable attention for its potential role in targeting BCR-ABL-positive leukemias. Experimental studies have shown that gallic acid inhibits BCR-ABL kinase activity and disrupts downstream oncogenic signaling, thereby reducing the proliferation and survival of CML cells. One of its mechanisms involves downregulation of cyclooxygenase-2 (COX-2) and inhibition of NF-κB activation, both of which contribute to leukemic cell growth and resistance to apoptosis [71]. Additionally, gallic acid has been reported to decrease the expression of matrix metalloproteinases MMP-2 and MMP-9, enzymes associated with invasion and metastasis, further highlighting its ability to suppress malignant progression in BCR-ABL-expressing leukemia cells [72]. Computational docking and molecular dynamics simulations provide additional evidence that gallic acid and its derivatives can bind effectively to the ATP-binding site of ABL kinase, supporting its role as a direct inhibitor of BCR-ABL activity [73]. More recently, gallic acid has also been shown to enhance the efficacy of TKIs, such as imatinib, by interfering with mitochondrial respiration and modulating oncogenic signaling networks in CML cells [74]. This synergistic effect suggests that gallic acid may not only act as a single agent but also function as an adjuvant that improves the therapeutic response to conventional BCR-ABL inhibitors.

Furthermore, gallic acid demonstrates rapid absorption following oral intake, with human studies showing detectable serum levels within an hour of consumption of green tea tablets, although the compound is also cleared quickly, resulting in limited systemic exposure and a relatively short half-life [75]. This low bioavailability is partly due to extensive metabolism and fast elimination, which can restrict its therapeutic potential. Strategies such as nano-delivery systems and formulation improvements have been investigated to enhance absorption and prolong circulation time [76]. Safety data from animal and human studies suggest that at typical dietary or moderate supplemental doses, gallic acid is generally well tolerated [77,78]. At pharmacological or higher doses, however, it may cause cytotoxic effects on normal cells, as reported in some preclinical models, raising the possibility of toxicity if administered in excess. Reported adverse effects include oxidative stress-related damage at high concentrations, gastrointestinal irritation, and possible interactions with drugs metabolized through overlapping pathways [76,79]. Overall, while gallic acid’s antioxidant and anticancer properties make it an attractive candidate for therapeutic use, its poor bioavailability and dose-dependent toxicities remain challenges. Improving delivery methods and establishing safe dosage ranges in clinical studies will be critical steps for translating its preclinical promise into therapeutic applications.

### 3.7. Artemisia Extracts

*Artemisia* extracts, particularly from *Artemisia vulgaris*, have gained attention for their potential therapeutic role in CML through direct effects on BCR-ABL signaling and related pathways. *Artemisia vulgaris* contains several biologically active compounds, including flavonoids, phenolic acids and sesquiterpene lactones. Among these, artemisinin (a sesquiterpene lactone endoperoxide) is recognized as the major bioactive constituent responsible for many of the plant’s pharmacological effects. A recent study demonstrated that *Artemisia vulgaris* extract significantly reduced BCR-ABL expression in CML cells, which was associated with inhibition of proliferation and induction of apoptosis. This suppression of BCR-ABL not only impaired leukemic cell survival but also enhanced apoptotic signaling, highlighting the extract’s potential to target the oncogenic driver of CML [80]. In parallel, artesunate, a semisynthetic derivative of artemisinin (*Artemisia annua*), has been shown to exert complementary antileukemic effects by inhibiting angiogenesis in CML cells. Specifically, artesunate decreased vascular endothelial growth factor (VEGF) expression in K562 cells, thereby impairing angiogenic support mechanisms that are critical for leukemic progression. It is important to note that artesunate is an FDA-approved antimalarial drug and not a direct natural compound from the plant extract [81]. The dual impact of *Artemisia*-based compounds, direct inhibition of BCR-ABL oncogenic signaling and suppression of angiogenesis, suggests a multifaceted mechanism of action that could complement TKIs. Importantly, both studies indicate that *Artemisia* derivatives trigger apoptosis selectively in leukemic cells, supporting their therapeutic potential while minimizing off-target effects [80,81]. *Artemisia vulgaris* extract in co-administration with imatinib has shown a synergistic anti-leukemic effect in CML cells. This combination significantly inhibited leukemia cell proliferation compared to treatment alone, enhanced apoptosis and reduced cell viability. These findings suggest that *Artemisia vulgaris* extract may potentiate the therapeutic efficacy of imatinib, offering a promising strategy to improve outcomes in CML treatment [82].

Human studies of *Artemisia* derivatives, particularly artesunate, show that certain formulations can be safe and well tolerated. In a trial comparing a new more stable parenteral formulation of artesunate to an existing one in healthy Thai volunteers, both intravenous and intramuscular versions were tolerated without new safety signals, and the drug was rapidly converted to its active metabolite, dihydroartemisinin, with high bioavailability for the metabolite after intramuscular administration [83]. Common adverse effects are generally mild and include transient changes in liver enzymes among others. Severe events are rare but post-artesunate delayed hemolysis has been observed in a small percentage of patients after treatment [84,85]. Regarding *Artemisia vulgaris* extract, there is some animal toxicity data, for example, hydrogels made from *Artemisia vulgaris* (and *Aloe vera*) showed no significant histological or biochemical abnormalities in liver, kidney, heart, or intestine in rodents after acute and sub-acute administration, though long-term human data are very sparse [86]. There is also evidence that *Artemisia vulgaris* extract can trigger precocious acrosome reaction and reduced sperm viability in vitro, suggesting potential reproductive toxicity under certain conditions [87].

A Phase II clinical trial is currently testing whether artesunate can improve outcomes when combined with imatinib in patients with chronic-phase CML. The study (NCT07022743) includes groups receiving imatinib alone, imatinib plus artesunate, and a cohort of patients who had prior exposure to imatinib. This trial is the first to explore the use of *Artemisia*-derived compounds together with standard therapy in CML patients, and its results will help clarify whether this combination can enhance responses or help overcome drug resistance [88].

### 3.8. Withaferin A

Withaferin A, a bioactive compound derived from *Withania somnifera*, has recently gained attention for its potential to target BCR-ABL signaling in CML. Computational analyses suggest that withaferin A directly interacts with the BCR-ABL oncoprotein, disrupting its kinase activity and downstream oncogenic signaling, including STAT5 and PI3K/Akt pathways that are critical for leukemic cell survival and proliferation [89]. By interfering with these pathways, withaferin A may suppress uncontrolled cell growth and enhance apoptotic responses in CML cells. Beyond computational predictions, accumulating pharmacological evidence highlights its broad anti-tumor activity, mediated by induction of oxidative stress, inhibition of chaperone proteins like HSP90, and modulation of transcription factors such as NF-κB [90]. Together, these findings position withaferin A as a promising candidate for overcoming resistance to TKIs in CML. However, experimental validation in CML-specific models remains necessary to translate these insights into therapeutic application.

However, its therapeutic use is limited by challenges in bioavailability and concerns regarding safety. Oral administration of withaferin A shows poor bioavailability, largely due to its low solubility and rapid metabolism, which reduces systemic drug exposure. Pharmacokinetic studies in mice demonstrated rapid clearance and low plasma concentrations following oral dosing, indicating that formulation strategies such as nanoparticle or liposomal delivery systems may be necessary to improve absorption. Despite these limitations, toxicity assessments in animal models suggest that withaferin A is well tolerated at high doses. In a comprehensive pharmacokinetic and safety evaluation in mice, oral administration of withaferin A up to 2000 mg/kg did not result in significant toxicological findings, indicating a wide margin of safety. Similarly, no major adverse effects on vital organs or hematological parameters were observed, further supporting its favorable safety profile. Current evidence therefore suggests that withaferin A is pharmacologically active but requires improved formulation strategies to overcome its limited bioavailability while maintaining its demonstrated safety at high oral doses [90,91].

## 4. Compounds Promoting BCR-ABL Degradation or Downregulation/Modulators of BCR-ABL Signaling

### 4.1. Triptolide

Triptolide, a diterpenoid triepoxide derived from the traditional Chinese medicinal plant *Tripterygium wilfordii*, has been widely studied for its strong antitumor effects, including activity against CML. One of its key mechanisms involves targeting the BCR-ABL oncogene. Unlike conventional TKIs that block the kinase activity of BCR-ABL, triptolide acts at the transcriptional level by suppressing BCR-ABL mRNA expression, leading to a marked reduction in protein levels. This downregulation not only inhibits the oncogenic signaling mediated by BCR-ABL, but also triggers apoptosis in CML cells, including those harboring the notoriously resistant T315I mutation [92]. Several studies have demonstrated that triptolide effectively induces apoptosis and inhibits proliferation in both imatinib-sensitive and imatinib-resistant CML cells. In K562/G01 cells, which exhibit resistance to imatinib, triptolide has been shown to disrupt survival signaling and restore apoptotic pathways, suggesting its potential as a therapeutic option when resistance develops [93]. Furthermore, synthetic water-soluble derivatives of triptolide have been designed to improve its pharmacological properties. These derivatives maintain strong antileukemic activity and effectively suppress the growth of T315I mutant CML cells, offering a promising direction for drug development [94]. The broader implications of triptolide in hematological malignancies also highlight its versatility. Its capacity to modulate multiple pathways, including transcriptional regulation and apoptosis induction, makes it a unique candidate compared to existing TKIs that focus narrowly on kinase inhibition. By targeting BCR-ABL expression rather than just its activity, triptolide provides an alternative strategy that may overcome resistance caused by mutations in the kinase domain [95]. Together, these findings position triptolide as a compelling compound with potential to complement or even extend current CML treatment strategies, particularly in cases where resistance to first- and second-generation TKIs limits therapeutic success. Triptolide in co-administration with imatinib has demonstrated a synergistic anti-leukemic effect in CML cells. This combination works by inhibiting BCR-ABL tyrosine kinase activity, leading to reduced cell proliferation and increased apoptosis in CML cell lines. Additionally, triptolide has been reported to enhance the efficacy of imatinib in overcoming drug resistance by modulating key signaling pathways involved in cell survival and proliferation [96].

However, its therapeutic use is significantly challenged by its poor water solubility and very limited oral bioavailability, which restrict its absorption and clinical potential [97]. Safety concerns further limit its application, and multiple studies have documented notable multi-organ toxicity, with the liver being particularly susceptible. Mechanistically, triptolide-induced hepatotoxicity appears to involve metabolic imbalance, oxidative stress, inflammation, autophagy, apoptosis, and interference with CYP450 enzyme activity and gut microbiota homeostasis [98]. Beyond the liver, triptolide and related compounds have demonstrated toxic effects on the kidneys, heart, reproductive system, and hearing, highlighting a broad toxicity profile that encompasses multiple organ systems [99]. These safety issues have prompted ongoing efforts to develop new delivery systems and derivative compounds, such as LLDT-8, PG490-88Na, and Minnelide, which aim to improve solubility, enhance bioavailability, and reduce systemic toxicity [100]. In summary, while triptolide offers powerful pharmacological actions, its suboptimal absorption and risk of significant adverse effects necessitate careful formulation strategies and dose optimization before any clinical translation can be realized.

### 4.2. Gambogic Acid

Gambogic acid, a natural compound derived from the resin of *Garcinia hanburyi*, has demonstrated significant inhibitory effects on BCR-ABL, particularly in CML cells resistant to conventional TKIs. Studies have shown that gambogic acid induces apoptosis in imatinib-resistant CML cells through multiple mechanisms, including proteasome inhibition and caspase-dependent downregulation of BCR-ABL protein levels [101]. By targeting BCR-ABL for degradation, rather than simply inhibiting its kinase activity, gambogic acid effectively suppresses the oncogenic signaling pathways that drive leukemic cell proliferation and survival. In addition to promoting apoptosis, gambogic acid has been reported to trigger autophagy in BCR-ABL-positive leukemia cells, suggesting that its anti-leukemic effects are mediated through both programmed cell death and autophagic processes [102]. Collectively, these findings suggest that gambogic acid holds promise as a therapeutic agent for BCR-ABL-driven leukemias, particularly in cases where resistance to first- or second-line TKIs has developed. Its multi-targeted mechanism of action, encompassing proteasome inhibition, BCR-ABL downregulation, apoptosis, and autophagy induction, positions gambogic acid as a potentially valuable compound for further preclinical and clinical investigation in CML.

Unfortunately, gambogic acid’s effect is hindered by low bioavailability and potential toxicity. Its poor aqueous solubility limits systemic absorption, necessitating the development of advanced delivery systems to enhance its therapeutic efficacy. Studies have demonstrated that gambogic acid’s bioavailability can be significantly improved through nanoformulations, leading to enhanced antitumor activity against leukemia cells in vitro and in vivo. This formulation also exhibited a higher half-lethal dose compared to gambogic acid’s water solution, suggesting reduced toxicity at effective doses [103]. Regarding safety, preclinical studies indicate that gambogic acid has a favorable safety profile when administered at appropriate doses. In rodent models, repeated oral doses of gambogic acid did not result in significant adverse effects, supporting its potential for safe use in humans [104]. Moreover, gambogic acid’s selective cytotoxicity towards tumor cells, coupled with its minimal impact on normal cells, underscores its therapeutic potential [105]. However, the clinical application of gambogic acid is still under investigation and further clinical trials are necessary to fully assess gambogic acid’s safety, efficacy, and optimal dosing regimens in CML treatment.

### 4.3. Celastrol

Celastrol, a quinone methide triterpene derived from *Tripterygium wilfordii*, has attracted attention for its ability to inhibit BCR-ABL signaling and overcome resistance in CML. Interestingly, it inhibits HSP90, a molecular chaperone that stabilizes BCR-ABL. By disrupting HSP90 function, celastrol promotes the degradation of BCR-ABL, including the imatinib-resistant T315I mutant, leading to apoptosis in CML cells [106]. Beyond targeting HSP90, celastrol also interferes with transcriptional regulation. It has been shown to inhibit YY1 and HMCES, proteins involved in DNA damage repair, which results in the accumulation of DNA damage and cell death in T315I-mutant CML cells [107]. This dual activity, destabilizing oncogenic proteins and impairing DNA repair, enhances celastrol’s ability to suppress resistant leukemic clones. In addition to its effects on BCR-ABL and DNA repair pathways, celastrol has been found to sensitize CML cells to chemotherapeutic agents. Studies show that when used in combination with other natural compounds, celastrol enhances apoptosis and reduces the survival of leukemia cells, highlighting its potential as a chemosensitizer [108]. This synergistic activity is particularly relevant in cases where monotherapy with TKIs fails due to acquired resistance. These findings suggest that celastrol, either as a single agent or in combination therapies, could represent a promising strategy for overcoming TKI resistance in CML and improving treatment outcomes.

Celastrol’s clinical use is limited by poor oral bioavailability and a narrow therapeutic window. Animal studies estimate that absolute oral bioavailability is relatively low due to poor solubility, rapid metabolism, and limited systemic absorption. Structurally, celastrol dissolves poorly in water, further restricting absorption and distribution [109,110]. Safety concerns arise from preclinical findings showing dose-dependent toxicity across multiple organs. High doses of celastrol have been associated with hepatotoxicity, nephrotoxicity, cardiotoxicity, and adverse effects on hematologic and reproductive systems. Additionally, its narrow therapeutic window can lead to toxic side effects even when only slightly exceeding effective doses [99,111]. To mitigate these challenges, researchers are exploring advanced drug delivery solutions. Nanotechnology-based methods, such as lipid nanospheres, liposomal encapsulation, polymeric micelles, and self-microemulsifying systems, have shown promise in enhancing celastrol’s solubility, improving oral bioavailability, and reducing off-target toxicity in preclinical models [112,113]. Taken together, while celastrol’s pharmacological effects are compelling, its practical application requires overcoming significant formulation hurdles and safety risks. Improving its bioavailability and therapeutic index via innovative delivery systems will be essential for translating preclinical successes into viable clinical therapies.

### 4.4. Andrographolide

Andrographolide, a diterpenoid lactone isolated from *Andrographis paniculata*, has been studied for its potential therapeutic role in CML particularly in the context of imatinib resistance. A key mechanism underlying its anticancer effect involves the suppression of the BCR-ABL oncoprotein. Studies have demonstrated that andrographolide effectively reduces BCR-ABL expression at the protein level, leading to the inhibition of downstream oncogenic signaling pathways and the induction of apoptosis in CML cells, including those resistant to TKIs [114]. Mechanistically, this effect is closely tied to the compound’s ability to disrupt molecular chaperone systems. Specifically, andrographolide promotes the cleavage of HSP90, thereby facilitating the degradation of this oncoprotein. In addition, the action of andrographolide is highly dependent on the generation of ROS, which serve as mediators of both HSP90 disruption and the suppression of cancer cell survival pathways [115]. Importantly, the development of andrographolide derivatives has further enhanced its potency. Modified analogs were shown to downregulate BCR-ABL more efficiently than the parent compound, with increased ability to trigger apoptosis in imatinib-resistant CML models [114]. To support mechanistic studies, bifunctional andrographolide-based probes have also been designed, allowing researchers to map molecular targets and clarify the compound’s pharmacological activities at a cellular level [116]. Collectively, these findings highlight andrographolide as a promising natural product with dual activity. It interferes with the stability of BCR-ABL while simultaneously generating intracellular stress signals that drive leukemic cell death. While these results remain preclinical, they provide a strong rationale for further investigation of andrographolide and its derivatives as potential therapeutic candidates against both imatinib-sensitive and -resistant CML.

Nevertheless, andrographolide faces challenges in clinical translation due to its poor oral bioavailability. This limitation stems from its low water solubility, rapid metabolism, including glucuronidation and sulfation, and active efflux by proteins like P-glycoprotein, which significantly reduce systemic absorption [117,118]. In a human pharmacokinetic study, maximum plasma concentrations of andrographolide plateaued even when the dose was doubled, highlighting issues with absorption and systemic exposure [119]. To address these issues, studies in animal models have explored the use of solubilizing agents and bioenhancers, for example, co-administering andrographolide with cyclodextrin or piperine has boosted its bioavailability by approximately 1.3 to 2 fold [120]. Safety data are encouraging given that an aqueous extract providing up to 360 mg/day of andrographolide over multiple doses was well tolerated in healthy volunteers, with only mild and transient adverse effects reported, like gastrointestinal discomfort, which promptly resolved [121]. At higher doses, however, there are occasional reports of hepatic enzyme elevation and hypersensitivity reactions, especially with injectable formulations. Preclinical studies suggest that andrographolide may cause nephrotoxicity and reproductive toxicity at high or prolonged doses. Animal data indicate potential kidney damage as well as reduced fertility [122]. These findings have not been confirmed in clinical settings, but they highlight the importance of careful dose optimization and monitoring in future studies.

### 4.5. Epigallocatechin-3-Gallate

Epigallocatechin-3-gallate (EGCG), the major polyphenolic compound in green tea, has attracted attention for its therapeutic potential in CML through its inhibitory action on BCR-ABL and related oncogenic pathways. Several studies have demonstrated that EGCG and its derivatives induce apoptosis in CML cells by suppressing BCR-ABL activity and downstream signaling. One important mechanism involves the activation of SHP-1, a tyrosine phosphatase that negatively regulates oncogenic kinases. By restoring SHP-1 activity, EGCG suppresses BCR-ABL and STAT3 signaling, thereby blocking cell survival pathways and inducing apoptosis in leukemic cells [123]. Another study revealed that EGCG triggers apoptosis through activation of acid sphingomyelinase, which elevates ceramide levels, disrupting membrane signaling and promoting programmed cell death [124]. EGCG also targets BCR-ABL-independent mechanisms of resistance, including sphingosine-1-phosphate signaling genes, which have been implicated in imatinib resistance. This suggests that EGCG may be useful not only in sensitive CML but also in resistant cases [125]. Furthermore, EGCG has been shown to modulate epigenetic regulation and senescence in leukemia cells, indicating broader effects on cellular reprogramming beyond kinase inhibition [126]. Importantly, EGCG demonstrates synergistic effects when combined with TKIs such as ponatinib, enhancing apoptosis and altering cell cycle regulatory gene expression, which strengthens its role as a potential adjuvant in therapy [127]. Additional studies highlight its regulation of key signaling cascades including p38-MAPK, JNK, JAK2/STAT3 and Akt, all of which are crucial mediators of CML cell proliferation and survival [128]. Taken together, these findings indicate that EGCG acts through multiple converging mechanisms to inhibit BCR-ABL and its downstream pathways, while also counteracting resistance mechanisms. This multifaceted activity underscores its promise as a complementary therapeutic candidate in CML management.

EGCG is absorbed into the bloodstream after oral intake of green tea or green tea extracts, but its bioavailability is modest. Peak plasma concentrations are usually reached within 1–2 h, followed by relatively rapid clearance. Because of this, many studies note that to maintain therapeutic levels, dosing or formulations, such as purified extracts, may need to be optimized [129,130]. In terms of safety, EGCG is generally well tolerated at doses commonly consumed in tea or standard supplements. However, higher doses or long-term use of concentrated extracts have been associated with liver-related adverse events in some human and animal studies, for example, elevated liver enzymes and, in rare cases, clinically significant liver injury [131,132]. Other adverse effects may include gastrointestinal symptoms [133]. Because EGCG interacts with many metabolic pathways, there is also potential for interactions with other drugs metabolized in the liver [131]. Overall, EGCG appears safe at moderate dietary or supplemental levels, but its modest bioavailability and risk of liver effects with high doses suggest caution, especially if considering it for adjunctive therapy in diseases like CML.

### 4.6. Curcumin

Curcumin, a polyphenolic compound derived from *Curcuma longa*, has been widely studied for its anticancer properties, including its potential to inhibit BCR-ABL signaling in CML. Several studies have demonstrated that curcumin exerts effects on BCR-ABL-expressing cells, leading to growth suppression and induction of apoptosis. In early reports, curcumin was shown to inhibit the STAT5 signaling pathway in primary CML cells, highlighting its ability to interfere with BCR-ABL downstream signaling cascades [134]. Importantly, curcumin has been effective against both wild-type and T315I-mutant BCR-ABL, inducing apoptosis and prolonging survival in mouse models of leukemia, suggesting its utility even in TKI-resistant disease [135]. Recent findings indicate that curcumin and its isoxazole analogs remain effective in imatinib-resistant CML cells, underscoring their therapeutic promise in overcoming drug resistance [136]. Combination studies also suggest that curcumin can act synergistically with other natural compounds enhancing apoptosis in CML cells [137]. Moreover, curcumin in co-administration with imatinib has been reported in vitro to more potently suppress proliferation and promote apoptosis of BCR-ABL-positive CML cells than either agent alone. For example, combined curcumin and imatinib reduced K562 cell growth and inhibited BCR-ABL signaling in cell-based studies [138,139]. Collectively, these findings highlight curcumin as a potent multi-target agent that inhibits BCR-ABL signaling, modulates oncogenic pathways, and holds promises for overcoming resistance in CML treatment.

Curcumin is known to have poor oral bioavailability due to low water solubility, rapid metabolism, and limited absorption in humans. Studies repeatedly show that after high oral doses, plasma levels of curcumin remain very low, and much of the administered compound is excreted unchanged or as metabolites in feces [140,141]. To improve this, several formulations have been developed (nanoparticles, liposomes, micelles, phospholipid complexes), which have shown higher absorption and better tissue distribution in animal and early human studies [142,143]. In terms of safety, curcumin is generally well tolerated in human trials even at high doses. Clinical studies administering up to 8 g per day in cancer patients reported minimal serious toxicity, with most adverse effects being mild and transient, such as digestive discomfort and mild gastrointestinal symptoms, and not requiring discontinuation. Some reports, however, note elevated liver enzymes at higher doses, and in rare cases liver injury in individuals using high-concentration curcumin supplements. Other side effects may include mild nausea, diarrhea, or headache. Because curcumin interacts with multiple metabolic and detoxifying pathways, there is potential for drug interactions, especially when highly bioavailable forms or high doses are used [142,144,145]. Overall, while curcumin appears safe at moderate to high doses with few severe adverse events, its low systemic exposure and possible dose-dependent liver effects mean that formulation strategies and careful dosing will be crucial for its use in human disease settings, especially those like CML where stable and effective drug levels are crucial.

A small randomized clinical study evaluated the effect of turmeric powder, a natural source of curcumin, as an adjunct to imatinib therapy in CML patients. The trial enrolled 50 patients with chronic-phase CML, who were randomized to receive either imatinib alone or imatinib in combination with turmeric powder (5 g/day, orally) for six weeks. The primary endpoint was the change in serum nitric oxide levels, used as a surrogate marker of oxidative stress and disease activity. The study reported that patients in the combination group had a greater reduction in serum nitric oxide levels compared to those receiving imatinib alone, suggesting a potential antioxidant and disease-modifying effect of turmeric. However, the study was limited by its small sample size, short duration, and lack of standardized curcumin content or clinical response endpoints. Despite these constraints, the study provides preliminary clinical evidence that curcumin-containing supplements may have a supportive role in modulating disease-related biochemical parameters in CML when combined with TKIs, though larger and more rigorously designed trials are needed to confirm efficacy and safety [146].

### 4.7. Resveratrol

Resveratrol, a naturally occurring polyphenol, has been extensively studied for its anti-leukemic activity in CML, particularly through its effects on BCR-ABL-dependent signaling. Early work demonstrated that imatinib-resistant CML cell lines remain highly sensitive to resveratrol, which suppresses cell proliferation and induces apoptosis, highlighting its potential to overcome resistance linked to BCR-ABL mutations [147]. One important mechanism involves the inhibition of STAT5, a critical downstream effector of BCR-ABL, where resveratrol induces the expression of SHP-1 and SHP-2 phosphatases, leading to reduced STAT5 phosphorylation and impaired transcription of survival genes [148]. Additional studies have shown that resveratrol activates stress-related pathways, such as p38 MAPK and JNK, resulting in phosphorylation of the DNA damage marker H2AX and subsequent apoptotic cell death [149]. Furthermore, resveratrol interferes with cell survival signaling by modulating Akt and ERK1/2 pathways, causing downregulation of HSP70, a molecular chaperone that stabilizes oncogenic proteins, including BCR-ABL [150]. Beyond direct molecular signaling, novel delivery strategies have also been explored, such as the use of resveratrol-loaded electrospun fibers combined with siRNA-liposomes, which enhance targeted cytotoxicity in CML co-culture systems and suggest translational potential for controlled release and synergistic therapies [151]. Resveratrol in co-administration with imatinib has shown enhanced anti-leukemic activity in CML cell models. Combined treatment produced greater inhibition of K562 cell growth and higher apoptosis rates than imatinib alone [152]. Taken together, these studies indicate that resveratrol exerts a multifaceted inhibitory effect on BCR-ABL-driven leukemogenesis, not by directly inhibiting the kinase, but by disrupting its downstream survival networks, activating pro-apoptotic stress responses, and modulating metabolic pathways. These findings position resveratrol as a promising adjunct agent for CML, particularly in cases of resistance to conventional TKIs.

Resveratrol is absorbed reasonably well in humans, but its bioavailability is low due to rapid metabolism in the intestines and liver, especially through conjugation and fast clearance from systemic circulation. In a pharmacokinetic study with 500 mg oral resveratrol tablets, about 75% of the dose was absorbed, yet most of what appears in the blood is in metabolite form rather than the intact parent compound. Peak plasma unmetabolized resveratrol levels are usually reached about 1.3 h after ingestion, with conjugated metabolites achieving higher and longer lasting concentrations [153]. To overcome these drawbacks, numerous technological and chemical strategies have been investigated to improve systemic bioavailability and stability. These include nanoformulations (polymeric nanoparticles, solid lipid nanoparticles, micelles, liposomes) that enhance intestinal absorption and protect resveratrol from degradation [154,155,156]. Additionally, prodrug approaches such as the synthesis of resveratrol derivatives and conjugates have demonstrated improved pharmacokinetic profiles and cellular uptake in preclinical studies. These advances collectively provide a foundation for developing clinically viable resveratrol-based therapeutics for leukemia and other malignancies [157,158,159]. In terms of safety, resveratrol is generally well tolerated at low to moderate doses. In the 500 mg study, healthy volunteers had minimal side effects. Most adverse events were mild, and no serious toxicity was observed. At higher doses (2–5 g/day), mild to moderate gastrointestinal symptoms such as nausea, diarrhea, and abdominal discomfort appear more frequently. There are occasional reports in the literature of potential drug interactions (via inhibition of CYP enzymes, platelet function), though definitive human data are limited [153,160]. Moreover, a recent clinical trial looking at a more bioavailable formulation of resveratrol confirmed that doses up to 1 g/day are tolerated with only mild adverse effects [161]. Still, the low systemic exposure of the parent compound, the high metabolism to conjugates, and the need for large doses to see measurable effects pose challenges for using resveratrol therapeutically in diseases like CML. Because of this, formulation improvements, dose schedules, and safety monitoring will be important for any future studies combining resveratrol with TKIs or other treatments.

### 4.8. Trifolium Repens

*Trifolium repens*, commonly known as white clover, has shown promising anticancer activity in CML by targeting oncogenic signaling. A study demonstrated that *Trifolium repens* extract inhibits proliferation of CML cells through suppression of the BCR-ABL/STAT5 pathway, reducing STAT5 phosphorylation and thereby downregulating genes essential for cell survival. This inhibition promoted apoptosis and impaired cell cycle progression, suggesting a direct impact on leukemic growth [162]. In addition, it has been reported that phytochemicals in *Trifolium repens*, such as flavonoids and phenolic compounds, contribute to its antiproliferative and antioxidant effects, supporting its multitargeted mechanism of action. These findings indicate that *Trifolium repens* may act as a natural modulator of BCR-ABL activity and could potentially complement TKI therapy [163]. However, further studies, particularly in vivo and clinical trials, are required to establish its therapeutic safety and efficacy in CML patients.

Current data on *Trifolium repens* safety and bioavailability are quite limited. Reviews of edible flowers of *Trifolium repens* note that white clover contains various phenolic compounds, flavonoids, and isoflavones, which contribute to antioxidant activity in vitro, but there are no human pharmacokinetic studies showing absorption, metabolism, or tissue distribution of these compounds. Animal or human safety data specific to anticancer use is scarce. The study of edible flower indicates that consumption of *Trifolium repens* in small amounts (as a food or garnish) is considered safe, with no reported adverse effects in those contexts [164,165]. Also, general reviews of safety of herbal products list *Trifolium* species among those with phytoestrogenic activity and possible interactions, but concrete reports of toxicity from standard consumption levels are minimal [165,166]. Because the specific doses used in experimental studies are generally much higher than dietary intake, the safety and adverse effect profile for therapeutic use remains largely uncharacterized. More research is needed to determine pharmacokinetics and long-term safety if moving toward clinical application in CML.

### 4.9. Xanthohumol

Xanthohumol, a prenylated flavonoid derived from hops, has been shown to interfere with BCR-ABL signaling in CML models. In BCR-ABL-positive K562 cells, xanthohumol induces caspase-dependent degradation of the BCR-ABL oncoprotein, leading to a marked reduction in cell viability and induction of apoptosis [167]. Earlier work also demonstrated that xanthohumol suppresses the growth of BCR-ABL-transformed cells through modulation of NF-κB and p53 pathways, suggesting that its anti-leukemic activity extends beyond direct effects on BCR-ABL protein stability [168]. Furthermore, xanthohumol enhanced the efficacy of imatinib by inhibiting imatinib-induced autophagy in K562 cells, suggesting potential for combination therapy to overcome resistance mechanisms associated with autophagy [167]. Together, these findings highlight xanthohumol as a natural compound capable of targeting BCR-ABL-driven leukemic cells via multiple mechanisms and support its further investigation as a potential adjunct in CML therapy.

Regarding its bioavailability, studies indicate that xanthohumol is absorbed after oral administration, with metabolites detectable in human plasma and urine, suggesting systemic exposure. However, its bioavailability is relatively low, which may limit its therapeutic efficacy. To enhance its bioavailability, various formulations and delivery systems are being explored [169,170]. Concerning safety, xanthohumol appears to be well-tolerated in humans. A phase I clinical trial administering 24 mg/day of xanthohumol over eight weeks reported no significant adverse effects, with participants showing no clinically relevant changes in liver function tests, complete blood counts, or electrolytes. Additionally, no serious adverse events were observed, and the compound did not negatively impact health-related quality of life measures [171]. Overall, xanthohumol holds promise as a therapeutic agent, but further research is needed to fully understand its pharmacokinetics, optimal dosing, and long-term safety profile.

## 5. Discussion

CML has been revolutionized by TKIs, yet resistance, particularly due to mutations like T315I, continues to limit therapeutic success. In this context, natural compounds represent a promising adjunctive approach, either by directly targeting BCR-ABL or by modulating signaling pathways that contribute to leukemic cell survival. Our review highlights a range of phytochemicals with potential inhibitory activity against BCR-ABL, grouped as direct or indirect modulators (Figure 7).

Direct inhibitors such as emodin, oridonin, berberine, chlorogenic acid, β-phenylethyl isothiocyanate, gallic acid, *Artemisia* extracts, and withaferin A, primarily act by directly interfering with BCR-ABLs kinase activity. These mechanisms offer an alternative route to overcome TKI resistance by destabilizing the oncoprotein rather than relying solely on catalytic inhibition [92,147]. Indirect inhibitors, including triptolide, gambogic acid, celastrol, andrographolide, EGCG, curcumin, resveratrol, and *Trifolium repens*, target upstream or downstream pathways such as NF-κB, STAT5, HSP90, and mitochondrial metabolism, thereby sensitizing leukemic cells to TKIs and reducing oncogenic signaling (Table 2) [106,172].

Despite the extensive preclinical evidence supporting the anti-leukemic effects of natural compounds against BCR-ABL, human clinical trials in CML remain remarkably scarce. Most investigations to date have been confined to in vitro studies and animal models, often demonstrating promising activity even in TKI-resistant cells. However, translation into clinical practice has been limited by poor bioavailability, uncertain safety profiles at therapeutic doses (Table 3), and potential drug–drug interactions with established TKIs. This translational gap between preclinical and clinical research is not unique to CML, as fewer than 5% of natural compound studies advance to human testing due to inconsistent standardization, low solubility and incomplete pharmacokinetic characterization. Similarly, while the volume of research on plant-derived anticancer agents has expanded significantly, their clinical validation remains disproportionately low, reinforcing the urgent need for integrated translational frameworks [173,174]. To bridge this gap, future research should prioritize well-designed clinical trials that assess the pharmacokinetics, optimal dosing strategies, and safety of these compounds, both as monotherapies and in rational combinations with TKIs [175].

Innovative drug delivery systems, such as nanoparticles, liposomes, or pro-drug formulations, could help overcome absorption barriers and enhance systemic availability. The use of sustainable, biocompatible nanocarriers derived from green chemistry principles has been proposed to enhance stability and bioefficacy while minimizing toxicity, thereby improving clinical applicability [176]. Furthermore, studies should not only evaluate efficacy against BCR-ABL but also investigate effects on resistance mechanisms, minimal residual disease, and leukemic stem cell persistence. In addition, integrating omics-based analyses with natural compound screening may help identify molecular targets and synergistic drug combinations that enhance anti-leukemic efficacy. Combinations of polyphenols with TKIs have been shown to potentiate apoptosis and inhibit proliferation in imatinib-resistant CML cells, underscoring their relevance as adjunct therapies [177,178]. Multicenter collaborative trials will be essential to establish standardized protocols and generate robust clinical evidence. Ultimately, integrating natural compounds into CML therapy will require a systematic approach that combines mechanistic insights with translational studies, moving beyond laboratory observations toward clinically meaningful outcomes [173,177].

In conclusion, natural products offer novel mechanisms to inhibit BCR-ABL or its signaling network, and several display synergistic activities with TKIs in resistant CML models. However, rigorous pharmacokinetic optimization and well-designed clinical trials are essential to determine their therapeutic viability. Moving forward, integrating phytochemicals with standard therapies could pave the way for improved outcomes in patients with resistant or advanced CML. This integration aligns with the emerging concept of sustainable, multi-targeted, and patient-tailored oncology, where natural compounds act not only as therapeutic agents but also as modulators of drug resistance, inflammation, and oxidative stress.

## Figures and Tables

**Figure 1 molecules-30-04160-f001:**
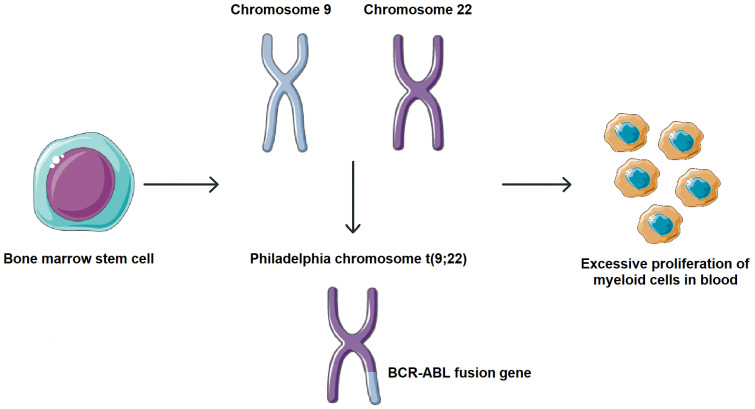
Molecular pathogenesis of CML. Formation of the BCR-ABL fusion gene. This cytogenetic aberration fuses the 5′ segment of the BCR gene on chromosome 22 with the 3′ segment of the ABL gene on chromosome 9, resulting in the formation of the BCR-ABL fusion gene. Parts of the image were provided by Servier Medical Art (https://smart.servier.com/), licensed under CC BY 4.0 (https://creativecommons.org/licenses/by/4.0/, accessed on 29 September 2025).

**Figure 2 molecules-30-04160-f002:**
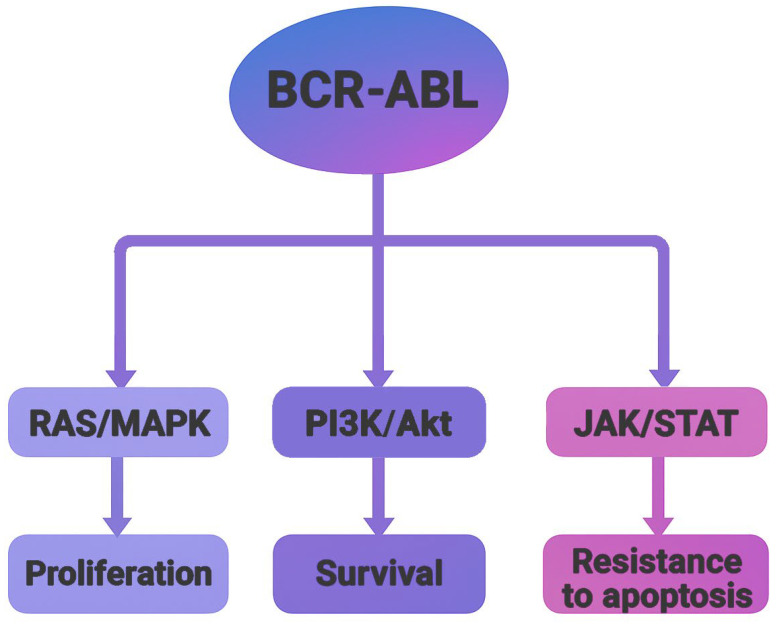
Schematic representation of major signaling pathways activated by the BCR-ABL oncoprotein. The fusion kinase activates RAS/MAPK, PI3K/Akt, and JAK/STAT pathways, leading to increased proliferation, survival and resistance to apoptosis.

**Figure 3 molecules-30-04160-f003:**
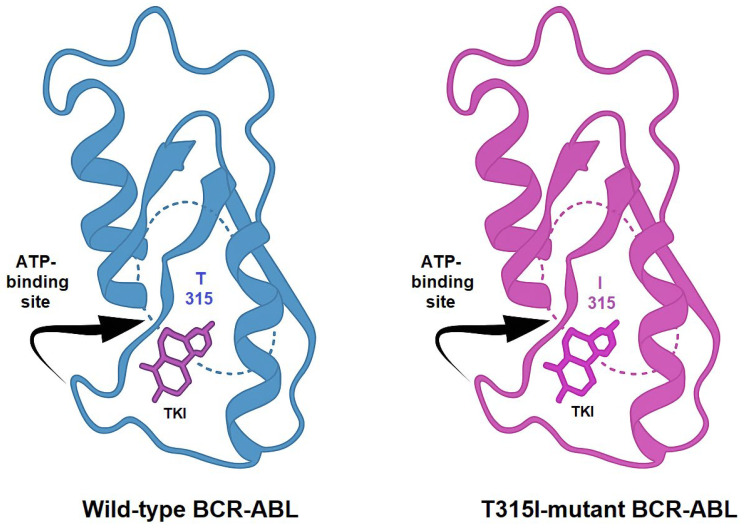
Comparative representation of the crystal structures of wild-type and T315I-mutated BCR-ABL kinase domains. The threonine-to-isoleucine substitution at residue 315 (T315I) reduces the binding affinity of first- and second-generation TKIs.

**Figure 4 molecules-30-04160-f004:**
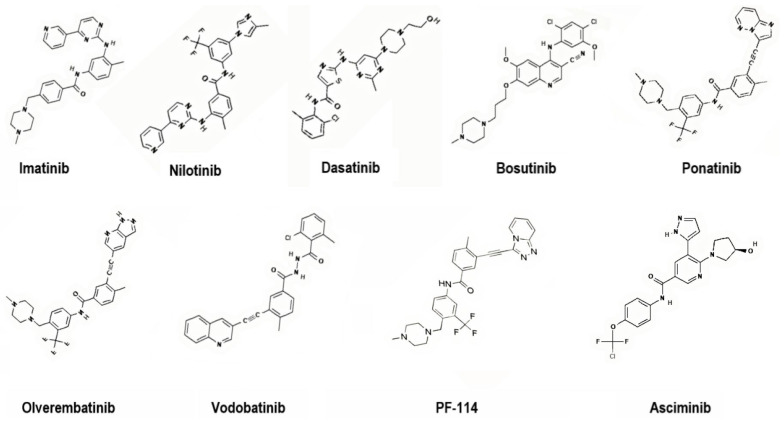
Representative chemical structures of first-, second-, third- and fourth-generation TKIs targeting BCR-ABL. 2D structure images were obtained from PubChem (PubChem Identifier CID for imatinib 5291; nilotinib 644241; dasatinib 3062316; bosutinib 5328940; ponatinib 24826799; olverembatinib 51038269; vodobatinib 89884852; PF-114 71475839; asciminib 72165228).

**Figure 5 molecules-30-04160-f005:**
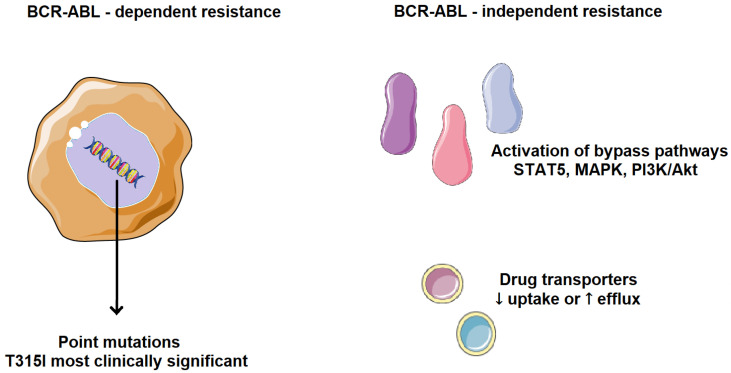
Mechanisms of TKI resistance in CML. BCR-ABL-dependent resistance is mainly influenced by the site of mutations in BCR-ABL and its uncontrolled gene expression. BCR-ABL-independent resistance mechanisms are mainly influenced by increased drug efflux, reduced drug uptake, and selective activation of tumor-promoting pathways. Parts of the image were provided by Servier Medical Art (https://smart.servier.com/), licensed under CC BY 4.0 (https://creativecommons.org/licenses/by/4.0/, accessed on 29 September 2025).

**Figure 6 molecules-30-04160-f006:**
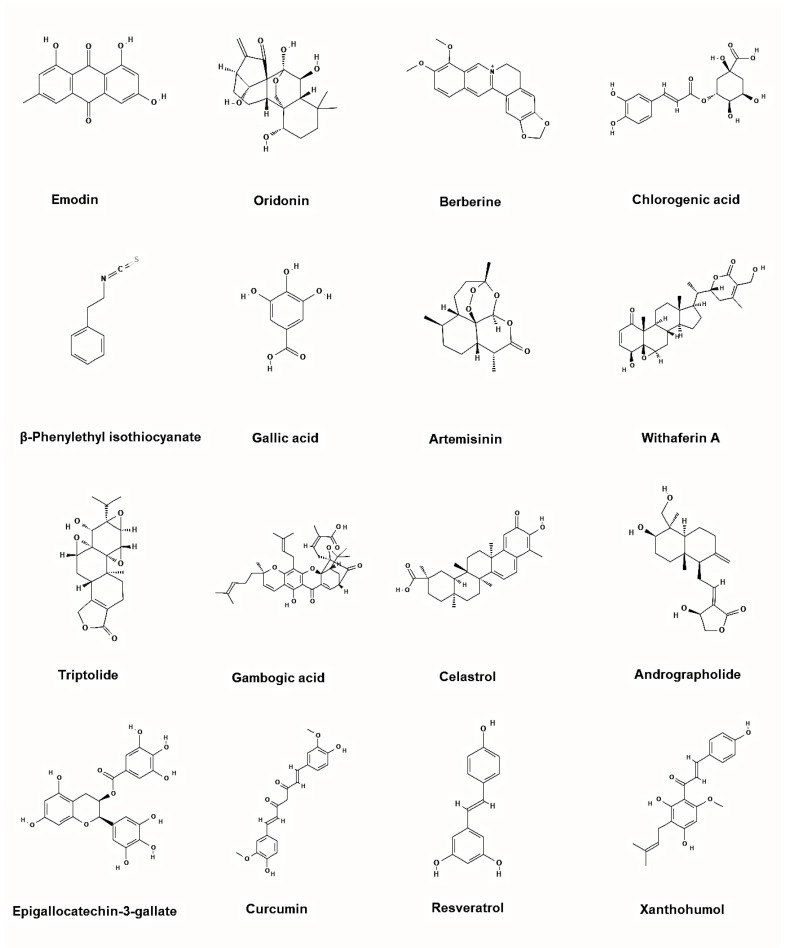
Representative chemical structures of the plant-based inhibitors of BCR-ABL. 2D structure images were obtained from PubChem (PubChem Identifier CID for emodin 3220; oridonin 5321010; berberine 2353; chlorogenic acid 1794427; β-Phenylethyl isothiocyanate 16741; gallic acid 370; artemisinin 68827; withaferin A 265237; triptolide 107985; gambogic acid 9852185; celastrol 122724; andrographolide 5318517; epigallocatechin-3-gallate 65064; curcumin 969516; resveratrol 445154; xanthohumol 639665).

**Figure 7 molecules-30-04160-f007:**
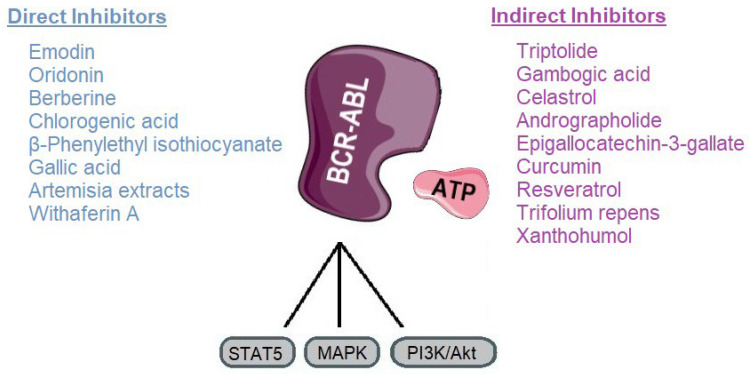
Classification of plant-derived compounds targeting BCR-ABL and its main signaling pathways. Parts of the image were provided by Servier Medical Art (https://smart.servier.com/), licensed under CC BY 4.0 (https://creativecommons.org/licenses/by/4.0/, accessed on 29 September 2025).

**Table 1 molecules-30-04160-t001:** Summary of pharmacological properties of clinically relevant BCR-ABL tyrosine kinase inhibitors. Most information and the NDA numbers were obtained by Drugs@FDA: FDA-Approved Drugs (https://www.accessdata.fda.gov/scripts/cder/daf/index.cfm, accessed on 18 October 2025).

TKI	Potency Scale	Bioavailability/ Absorption	Apparent Half-Life	Major Toxicities/Safety Concerns
Imatinib (NDA #021335)	Low/ moderate	High	~18 h	Edema, hepatotoxicity, cytopenias, gastrointestinal disturbances
Nilotinib (NDA #022068)	Moderate	Moderate	~15–17 h	QT prolongation, metabolic disturbances, hepatotoxicity
Dasatinib (NDA #021986)	High	Low/ moderate	~3–5 h	Pleural effusion, cytopenias, bleeding, QT prolongation
Bosutinib (NDA #203341)	High	Moderate	~22.5 h	Diarrhea, hepatotoxicity, cytopenias
Ponatinib (NDA #203469)	Very high	Moderate	~24 h	Arterial occlusive events, pancreatitis, hepatotoxicity, hypertension
Olverembatinib [21,22,23]	High	Moderate	~32.7 h	Cardiovascular events, hypertension, pericardial effusion, cytopenias, elevated CPK, thrombocytopenia
Vodobatinib [24,25]	Moderate/ high	Moderate	Not well characterized	Safety profile in trials is favorable to date (less off-target toxicity)
PF-114 [26,27]	High	Moderate	~13.5 h	Skin hyperpigmentation, proteinuria, elevated liver enzymes, hypertriglyceridemia
Asciminib (NDA #215358) [28]	High/ very high	High	~7 to 15 h	Fatigue, nausea, headache, diarrhea, elevated liver enzymes, thrombocytopenia, neutropenia, rare QT prolongation

**Table 2 molecules-30-04160-t002:** Plant-derived compounds targeting BCR-ABL, their main mechanism of action and key molecular targets.

Compound	Classification	Main Mechanism of Action	Key Molecular Targets
Emodin	Direct inhibitor	Allosteric inhibition and downregulation of BCR-ABL expression.	BCR-ABL, STAT5
Oridonin	Direct inhibitor	Promotes proteasome-mediated degradation of BCR-ABL.	BCR-ABL, HSF1-HSP70 axis
Berberine	Direct inhibitor	Autophagy-dependent degradation of wild-type and T315I BCR-ABL.	BCR-ABL, LRSAM1, ubiquitin pathway
Chlorogenic acid	Direct inhibitor	Inhibits BCR-ABL kinase activity, induces apoptosis via ROS.	BCR-ABL, p38 MAPK
β-Phenylethyl isothiocyanate	Direct inhibitor	Generates ROS, disrupts PKC-BCR-ABL crosstalk, induces apoptosis.	BCR-ABL, PKC, ROS pathway
Gallic acid	Direct inhibitor	Inhibits BCR-ABL phosphorylation, downregulates COX-2, NF-κB.	BCR-ABL, NF-κB, COX-2
Artemisia extracts	Direct inhibitor	Reduces BCR-ABL expression and promotes apoptosis.	BCR-ABL, VEGF signaling
Withaferin A	Direct inhibitor	Predicted binding to BCR-ABL, suppresses oncogenic signaling.	BCR-ABL, JAK/STAT, PI3K/Akt
Triptolide	Indirect inhibitor	Inhibits BCR-ABL transcription, induces apoptosis in resistant cells.	BCR-ABL mRNA, NF-κB
Gambogic acid	Indirect inhibitor	Proteasome inhibition leading to BCR-ABL downregulation.	BCR-ABL, caspases
Celastrol	Indirect inhibitor	Disrupts HSP90 function, destabilizing BCR-ABL.	HSP90, BCR-ABL
Andrographolide	Indirect inhibitor	Induces ROS, HSP90 cleavage, reduces BCR-ABL signaling.	HSP90, ROS pathway
Epigallocatechin-3-gallate	Indirect inhibitor	Activates SHP-1 phosphatase, suppresses BCR-ABL/STAT3.	BCR-ABL, STAT3, SHP-1
Curcumin	Indirect inhibitor	Inhibits STAT5 and exosomal miRNA signaling, induces apoptosis	STAT5, NF-κB, miR-21
Resveratrol	Indirect inhibitor	Suppresses STAT5, Akt/ERK pathways, downregulates HSP70.	STAT5, PI3K/Akt, ERK1/2, HSP70
Trifolium repens	Indirect inhibitor	Suppresses BCR-ABL/STAT5 signaling, inhibits proliferation.	BCR-ABL/STAT5 pathway
Xanthohumol	Indirect inhibitor	Induces caspase-dependent degradation of BCR-ABL, suppresses NF-κB and p53 signaling.	BCR-ABL, NF-κB, p53, autophagy pathway

**Table 3 molecules-30-04160-t003:** Bioavailability and toxicity profile of plant-derived compounds targeting BCR-ABL.

Compound	Bioavailability	Toxicity Profile
Emodin	Poor oral bioavailability due to limited absorption and rapid metabolism.	Generally safe at low doses. High doses associated with hepatotoxicity and nephrotoxicity in preclinical studies.
Oridonin	Low solubility and poor pharmacokinetics. Derivatives are being developed to improve bioavailability.	It can cause hepatotoxicity and gastrointestinal toxicity at higher doses.
Berberine	Very low oral bioavailability (<1%). Affected by P-glycoprotein efflux.	Safe at moderate doses. Gastrointestinal discomfort and potential hepatotoxicity at high doses.
Chlorogenic acid	Limited stability and variable oral absorption.	Considered safe. Excessive intake may cause gastrointestinal effects.
β-Phenylethyl isothiocyanate	Moderate bioavailability from dietary sources.	Safe at nutritional levels. High doses may cause gastrointestinal irritation.
Gallic acid	Low oral bioavailability due to rapid metabolism.	Safe at dietary levels. Possible oxidative stress at high doses.
Artemisia extracts	Variable bioavailability. Artesunate more stable.	Safe at therapeutic doses. Possible hepatotoxicity and neurotoxicity at high levels.
Withaferin A	Low oral bioavailability. Limited pharmacokinetic data.	Safe up to 2000 mg/kg in mice. Higher doses may cause hepatotoxic or reproductive effects.
Triptolide	Very low oral bioavailability.	Narrow therapeutic window. Hepatotoxicity, nephrotoxicity, and reproductive toxicity reported.
Gambogic acid	Poor solubility and low oral absorption.	Selective cytotoxicity toward tumor cells. Hepatotoxicity and gastrointestinal effects at higher doses.
Celastrol	Limited oral bioavailability. Highly lipophilic.	Hepatotoxicity, nephrotoxicity, and reproductive toxicity in preclinical models.
Andrographolide	Low oral bioavailability.	Generally safe at clinical doses. High doses are linked to hepatotoxicity and reproductive toxicity.
Epigallocatechin-3-gallate	Poor oral bioavailability due to rapid metabolism.	Safe in dietary amounts. Hepatotoxicity observed in concentrated supplements.
Curcumin	Extremely low oral bioavailability (<1%). Enhanced by piperine or formulations.	Very safe. Mild gastrointestinal or hepatotoxic effects at high doses.
Resveratrol	Very low oral bioavailability. Rapid metabolism.	Safe at moderate doses. Gastrointestinal upset at high doses.
Trifolium repens	Limited bioavailability data. Similar to other isoflavonoids.	Considered safe at dietary levels. Possible estrogenic effects with excessive use.
Xanthohumol	Absorbed after oral administration. Overall bioavailability is relatively low.	Appears well-tolerated in humans. No significant adverse effects.

## Data Availability

No new data were created or analyzed in this study. Data sharing is not applicable to this article.

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
