# Peer review of "Natural Products Targeting BCR-ABL: A Plant-Based Approach to Chronic Myeloid Leukemia Treatment"

_molecules, 2025, doi:10.3390/molecules30214160_

Round 1

Reviewer 1 Report

Comments and Suggestions for Authors

This review article aims to highlight the importance of plant-derived agents as useful adjunctive therapies against BCR-ABL-driven leukemias. The review is well written, and the authors clearly summarized the effects of selected plant-derived compounds targeting BCR-ABL according to their direct or indirect modulation towards the target. The major criticism is the lack of informative figures about BCR-ABL-mediated downstream pathways and the chemical structures of reported compounds that are missing. Despite this critical aspect that needs to be addressed, I believe that the manuscript may be of general interest to the readers of this journal, hence it deserves to be published after minor revision.

My own major suggestions are as follows:

  • Introduction (page 2, line 62): to increase the overall quality of the manuscript, the authors should provide an informative and schematic figure that illustrates the multiple BCR-ABL activated pathways described in the text.
  • Introduction (page 2, line 68): a figure comparing the crystal structures of wildtype versus mutated BCR-ALB protein should be provided to give a molecular insight into the structural determinants for selective targeting of mutated protein by novel generation of TKIs.
  • Chemical structures and related inhibitory potency of cited TKIs should be provided in the articles. Similarly, the chemical structures for all natural compounds described in this manuscript should be reported. I believe that many chemists will appreciate finding those pieces of information in the article.
  • Page 4, line 137: a table summarizing the overall ADME-Tox properties, as well as metabolic pathways and drug-drug interactions, of the most relevant TKIs would be very valuable.
  • Page 9, lines 357 and 378: since the author reported a plant extract, it would be better to specify the major components in terms of compound content. For instance, is artemisinin the major one? If so, define its nature (i.e., sesquiterpene lactones). Also, the authors should specify that artesunate is an FDA-approved drug clinically used to treat malaria; otherwise, it seems like it is a natural plant derivative rather than a semisynthetic drug.
  • Page 16, lines 706-708: since low oral bioavailability of resveratrol is the major drawback that limits its clinical use, the authors should consider mentioning the many technological (drug delivery system) and chemical strategies (prodrugs) useful to improve resveratrol systemic bioavailability by providing relevant references.
  • Discussion: The author should consider expanding this section by providing current trends and perspectives for the future direction of the research in the field of plant-based approaches to treat CML.

Minor suggestions:

  • Introduction (page 3, lines 73, 74): As a general rule, nonproprietary names begin in lowercase, while trade names begin with a capital. Please be consistent with the text and correct the names accordingly.
  • Page 5, line 185: carefully revised the formatting of the text to improve consistency. As “Rabdosia rubescens” is written in plain type, while should be in italics. Please carefully review the text for typos.
  • Page 7, line 259 and then line 260 (i.e., CGA): similarly, spell out most acronyms on first use and define them at first mention. Please be consistent within the text.
  • Line 288: Correct the spelling; an extra comma needs to be removed.

Reviewer 2 Report

Comments and Suggestions for Authors

In this manuscript, the author did a comprehensive overview of natural compounds with potential anti-leukemic activity through modulating BCR-ABL signaling. The authors successfully integrated molecular pharmacology and medicinal chemistry perspectives and presented clear mechanism for most compounds. The manuscript is recommended for publication with the following minor revisions:

  1. While the review is comprehensive, it sometimes reads as a list of compounds without deeper comparative or critical synthesis. It is recommended that the authors add a summary table or figure comparing mechanisms of action, bioavailability, toxicity, and synergistic potential of the key natural products to facilitate reader understanding.
  2. It would be clearer for comparing each compound if the authors explicitly classify evidence levels (e.g., preclinical, in vivo, or clinical) to avoid overstating conclusions.
  3. It would be better to show chemical structures of compounds mentioned or heavily discussed.
  4. The concept "indirect inhibitor" is a bit strange as most inhibitors will indirectly inhibit the downstream pathways. It is recommended that the authors reduce the discussion in this part to avoid confusion.
